# Evaluating the impact of sex bias on AI models in musculoskeletal ultrasound of joint recess distension

M. Mendez[1], N. Jafarpisheh[1], S. Demello[1], C. Lee[1], M. Dang[1], P. N. Tyrrell[1,2,3]*

1 Department of Medical Imaging, University of Toronto, Toronto, Ontario, Canada, 2 Institute of Medical Sciences, University of Toronto, Toronto, Ontario, Canada, 3 Department of Statistical Sciences, University of Toronto, Toronto, Ontario, Canada

* pascal.tyrrell@utoronto.ca

## Abstract

With the increasing integration of artificial intelligence (AI) in healthcare, concerns about bias in AI models have emerged, particularly regarding demographic factors. In medical imaging, biases in training datasets can significantly impact diagnostic accuracy, leading to unequal healthcare outcomes. This study assessed the impact of sex bias on AI models for diagnosing knee joint recess distension using ultrasound imaging. We utilized a retrospective dataset from community clinics across Canada, comprising 5,000 de-identified MSKUS images categorized by sex and clinical findings. Two binary convolutional neural network (BCNN) classifiers were developed to detect synovial recess distension and determine patient sex. The dataset was balanced across sex and joint recess distension, with models trained using advanced data augmentation and validated through both individual and mixed demographic scenarios using a 5-fold cross-validation strategy. Our BCNN classifiers showed that AI performance varied significantly based on the training data's demographic characteristics. Models trained exclusively on female datasets achieved higher sensitivity and accuracy but exhibited decreased specificity when applied to male images, suggesting a tendency to overfit female-specific features. Conversely, classifiers trained on balanced datasets displayed enhanced generalizability. This was evident from the classification heatmaps, which varied less between sexes, aligning more closely with clinically relevant features. The study highlights the critical influence of demographic biases on the diagnostic accuracy of AI models in medical imaging. Our results demonstrate the necessity for thorough cross-demographic validation and training on diverse datasets to mitigate biases. These findings are based on a supervised CNN model; evaluating whether they extend to other architectures, such as self-supervised learning (SSL) methods, foundation models, and Vision Transformers (ViTs), remains a direction for future research.

**Data availability statement:** The data underlying the findings of this study cannot be made publicly available due to confidentiality and data transfer agreements with the data provider. The restricting institution is the University of Toronto. Researchers interested in accessing these data may submit a request. Access will be granted upon reasonable request to broaden this research, subject to ethical approval and any additional data-sharing agreements required by the institution. Requests should be directed to the corresponding author and the University of Toronto Research Ethics Board at ethics.review@utoronto.ca (RIS protocol #39361).

**Funding:** This work was supported by Novo Nordisk Health Care AG, grant #2020-0922. Granted to PNT.The funders had no role in study design, data collection and analysis, decision to publish, or preparation of the manuscript.

## Introduction

Synovial recess distension, characterized by the swelling of the synovial-lined recesses in joints such as the knee, occurs due to fluid accumulation, often triggered by inflammation, overuse, or injury [1]. Primarily affecting adults engaged in physically demanding activities, this condition not only significantly impairs mobility and quality of life but also leads to complications such as joint stiffness, pain, and reduced range of motion, which collectively burden the healthcare system through high prevalence and the chronic nature of joint disorders, necessitating long-term management strategies [2,3].

Musculoskeletal ultrasound (MSKUS) is used for the diagnosis of synovial recess distension, offering a non-invasive, cost-effective, and readily available method to visualize soft tissue structures. Ultrasound facilitates early detection and assessment of the extent of inflammation [4,5], aiding in timely and accurate diagnosis which is crucial for effective treatment planning. Despite its advantages in dynamic joint function assessment and ability to perform comparative assessments with the contralateral limb, it is often complemented by MRI, which provides a definitive evaluation of the joint structures [6, 7].

The integration of artificial intelligence (AI), including convolutional neural networks (CNNs), has significantly advanced early diagnosis in medical imaging [8–10]. CNNs have shown promise in automating the diagnosis of musculoskeletal conditions by analyzing ultrasound images. The general pipeline for AI-assisted diagnosis of synovial recess distension follows a structured approach: (1) preprocessing of ultrasound images to enhance relevant anatomical structures, (2) feature extraction using deep learning models, and (3) classification based on learned representations distinguishing between normal and distended synovial recesses. Previous research has demonstrated that CNNs can effectively identify fluid accumulation in synovial recesses, aiding in the early detection of inflammatory conditions such as arthritis. However, these models are susceptible to dataset biases, particularly those related to patient demographics such as sex, which can impact diagnostic accuracy and generalizability. Despite advancements in AI for musculoskeletal ultrasound, there remains a significant gap in understanding how demographic factors influence model performance. This study aims to address this gap by systematically evaluating the effects of training data composition on AI-driven classification of synovial recess distension.

AI algorithms can analyze ultrasound images with high precision [11–14], potentially identifying subtle patterns of synovial recess distension that may be overlooked in manual examinations. However, while AI promises enhanced accuracy and efficiency, it also introduces potential disadvantages such as algorithmic opacity and dependency on the quality of the data used for training these models.

A significant yet often overlooked pitfall in the application of AI in medical imaging is the bias inherent in training datasets. Biases, both implicit and explicit, can skew AI performance, leading to misdiagnoses or systematic disparities in healthcare delivery [15,16]. The assessment of bias, particularly in models trained with demographic-specific data, is critical yet frequently neglected in research [17]. The

assumption that AI models will inherently overcome bias with sufficient data is a common misconception that overlooks the fundamental issues present in the data itself.

Research has shown that ultrasound images exhibit variations across demographics such as sex, age, and ethnicity, underscoring potential biases in image interpretation [18]. While extensive literature explores bias in general AI applications [19–22], investigations specifically within AI-assisted MSKUS diagnosis are notably less frequent. This underrepresentation underscores the necessity of our study, which aims to sensitize the medical community to the subtle yet significant impacts of dataset biases on AI-driven diagnostics using MSKUS as imaging modality.

A growing body of clinical research highlights significant sex differences in musculoskeletal anatomy and osteoarthritis (OA). Systematic reviews and clinical studies have documented that women often experience a higher prevalence of OA, along with distinct morphometric differences in joint structures, compared to men [23]. These differences are attributed to a complex interplay of hormonal, genetic, and biomechanical factors, which may render female joints more vulnerable to degenerative changes. Moreover, emerging evidence suggests that AI models trained on unbalanced datasets might inadvertently reinforce these biases, leading to disparate diagnostic outcomes across sexes [24]. Overall, these studies not only provide a critical clinical context for understanding the impact of sex on OA but also highlight the necessity for balanced, demographically representative training datasets to ensure equitable performance in AI-driven diagnostics.

Specifically, this study seeks to quantitatively assess how sex-related biases affect the performance of AI models used in diagnosing knee recess distension via MSKUS. Often, due to oversight or data availability, models may be inadvertently trained on datasets skewed towards a single sex, potentially biasing their clinical applicability. By examining the performance differences between models trained on male-only, female-only, and balanced datasets, this research aims to demonstrate the necessity for and benefits of demographic curation in training data. The goal is to ensure that AI diagnostics perform equitably and effectively across diverse patient demographics, ultimately enhancing clinical reliability and patient care.

While ViTs introduce self-attention mechanisms that enhance feature extraction, the effect of dataset bias on ViTs is expected to be similar to that on CNNs—model fairness and generalization are primarily dictated by the composition and diversity of the training data, rather than architecture choice. CNNs, particularly architectures like EfficientNet, remain widely used in real-world medical settings because of their computational efficiency, established performance, and ability to generalize well on moderate-sized datasets [25,26].

Fig 1 provides a comprehensive overview of the study workflow, outlining the key stages from data preparation to model evaluation and bias analysis. The pipeline begins with dataset preparation, including expert screening and annotation, followed by data partitioning and cross-validation. The model training phase employs EfficientNet-B4 across three training scenarios (Male-Only, Female-Only, and Balanced), incorporating data augmentation strategies to enhance generalizability. Finally, model performance is evaluated through multiple metrics, including accuracy, sensitivity, specificity, and AUC, with additional bias analysis using Grad-CAM visualization to interpret model decisions across different training conditions. This structured approach ensures a thorough investigation of demographic biases in AI-driven musculoskeletal ultrasound diagnosis.

The remainder of this paper is structured to first detail the methods used for data collection and AI model training in Section 2. Section 3 will then present a thorough analysis of the results, followed by a discussion in Section 4 on the implications of our findings for the future of AI in MSKUS. Finally, Section 5 will provide concluding remarks to synthesize our study's key insights.

## Materials and methods

### Materials

This retrospective study capitalized on a dataset comprised of de-identified MSKUS images sourced from an array of community clinics in Ontario, Canada, focusing exclusively on adult patients undergoing knee joint examinations. The

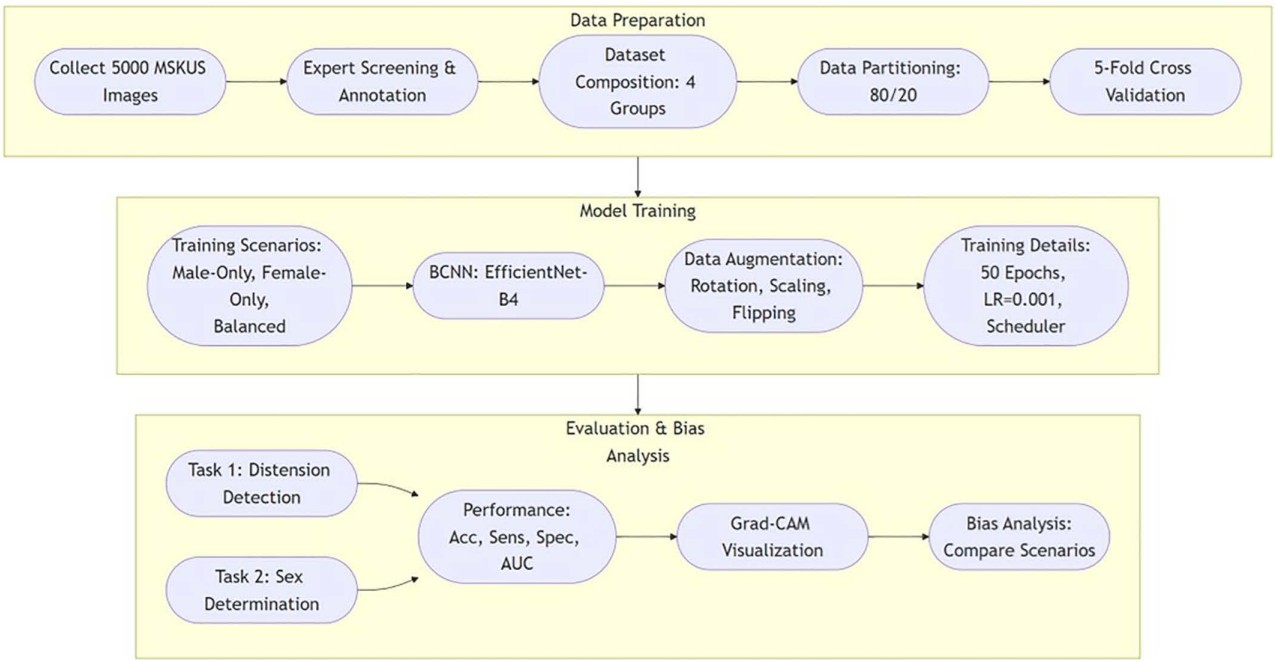

**Fig 1. Overview of the study workflow for AI-driven musculoskeletal ultrasound analysis.** The pipeline consists of three main stages: (1) Data Preparation, including expert screening, dataset composition, and partitioning; (2) Model Training, utilizing EfficientNet-B4 with different training scenarios and data augmentation strategies; and (3) Evaluation & Bias Analysis, which involves performance assessment, Grad-CAM visualization, and bias comparison across training conditions.

study adhered to all applicable ethical guidelines, with data acquisition (during September 2021) and usage approved under RIS protocol #39361.

**Dataset composition.** Ultrasound examinations were conducted between January 1, 2010, and December 31, 2018, targeting patients with an average age of 50 years. These images, centered around the evaluation of the knee joint's synovial recess, were screened to ensure only those showcasing the suprapatellar view were included. The final dataset, carefully curated to ensure both sex and clinical condition representation, included 5,000 images across four patient cohorts: male patients with synovial recess distension, male patients without synovial recess distension, female patients with synovial recess distension, female patients without synovial recess distension. Each subgroup contributed 1,250 images, allowing balanced comparative analyses. All patient information was stripped from the records except for necessary demographic details (sex) and clinical findings (synovial recess distension), complying with stringent privacy standards.

**Image review and validation.** All images underwent a preliminary screening process conducted by an expert sonographer with extensive experience (32 years) to validate the presence or absence of synovial recess distension. This rigorous review ensured that the dataset was reliably annotated, thus strengthening the integrity of subsequent analyses.

**Data handling and partitioning.** To simulate real-world diagnostic applications, the dataset was partitioned into training (80%) and testing (20%) sets. The training data underwent further division, employing a 5-fold validation strategy to enhance the metrics of the model's performance evaluation. This division was meticulously designed to maintain an equal distribution of the four distinct patient cohorts in each fold, ensuring comprehensive model training and validation.

**AI detection of synovial recess distension**

In this study, two binary convolutional neural network (BCNN) classifiers were developed to address distinct tasks using knee ultrasound imaging: detecting synovial recess distension of the knee and determining patient sex. The model architecture, training protocol, and hyperparameters were standardized across both classifiers to facilitate consistent performance evaluation.

**Model development and training.** Both classifiers were based on the EfficientNet-B4 architecture [27], chosen for its efficiency and strong performance in image classification tasks. The models were trained using a comprehensive data augmentation strategy—including techniques such as rotation, scaling, and flipping—to increase the diversity of the training set and improve the models' ability to generalize to new, unseen images.

The training regimen involved 50 epochs, starting with a learning rate of 0.001 that was adjusted via a decreasing scheduler to optimize convergence. The models utilized binary cross-entropy loss with the Adam optimizer to guide the training process. Considering the extensive dataset, a batch size of 64 was selected for optimal use of the computational resources, specifically a Nvidia 4090 GPU, enabling efficient processing of large-scale image data. To ensure reproducibility and transparency, we acknowledge the importance of providing access to the source code. While data cannot be publicly shared due to ethical and privacy considerations, we have made the core implementation of our model, preprocessing pipeline, and evaluation methodology available in https://github.com/Noushh/PLOS-One-paper.

Initialization of the model weights was from a pre-trained state on the ImageNet dataset, leveraging transfer learning to enhance initial performance and accelerate convergence.

**Model validation and performance assessment.** To assess the impact of training with a biased dataset on the detection performance of synovial recess distension, a series of structured assessments were conducted. These evaluations focused on models trained exclusively on data from a single sex (either male or female cohorts), as well as on models trained on a combined dataset. Each scenario was carefully crafted to reflect the same distribution of images to maintain consistency (50% positive for synovial recess distension).

Additionally, the performance of the classifiers in detecting patient sex was evaluated using a balanced set of images from both sexes, with and without recess distension, to ensure that the models were not biased by the underlying condition of the knee.

Throughout these evaluations, the models were assessed using metrics such as accuracy, sensitivity, specificity, and area under the receiver operating characteristic curve (AUC). These metrics provided a multi-dimensional view of model performance, highlighting strengths and identifying potential areas for improvement in handling clinical imaging data. Classification heatmaps, using Grad-CAM [28], were calculated to further explain the bias impact on the models.

To provide a more robust statistical evaluation, we employed a paired bootstrapping analysis on a fixed, balanced test set [29]. The test set remained constant across all experimental conditions, ensuring equal representation of male and female images. For each pairwise comparison, we performed 5,000 iterations of bootstrapping, where we resampled the test set with replacement and computed the accuracy difference between models at each iteration. From these resampled distributions, we calculated the mean accuracy difference, standard deviation, and 95% confidence intervals, along with p-values to assess statistical significance. This method ensures a statistically rigorous evaluation of how demographic bias in training data affects model performance across three training scenarios: (1) Male-Only, (2) Female-Only, and (3) Balanced Training. This approach ensures a statistically sound assessment of demographic bias effects on model performance.

## Results

The analysis of BCNN classifiers for detecting synovial recess distension and identifying patient sex may offer important understanding of the impacts of dataset bias in developing AI tools for knee ultrasound diagnosis.

## Comparative analysis of joint recess distension model training on sex-specific data

Table 1 provides the outcome from training the BCNN models on datasets segregated by sex (male, female, or both), tested across a sex-balanced dataset. The model trained on female data showed a higher accuracy (87.47%) and sensitivity (88.46%) compared to those trained solely on male data (accuracy of 83.98% and sensitivity of 83.11%), suggesting that features pertinent to female anatomy or pathology may be more distinct or consistently imaged than those in males. When models were trained on both male and female datasets, there was a slight improvement in accuracy (87.73%) and sensitivity (88.51%), with a specificity of 84.29%, slightly higher than that seen with the female-only training set (83.10%), indicating the benefit of a more heterogeneous training set in enhancing model robustness. The AUC values remained high across all training settings, indicating strong discriminative ability, with the highest noted in the female-only training scenario (0.9269).

Exploring deeper, Table 2 highlights the performance across sex-specific test subsets, revealing nuanced differences. Models trained on female-only data not only excelled in identifying conditions in females (accuracy 87.87%, sensitivity 87.73%) but also performed reasonably well with male-only test data (accuracy 87.42%, sensitivity 89.61%). However, there was a noticeable drop in specificity for males (76.92%) when the model was trained on only females, suggesting potential overfitting or bias towards identifying features more prevalent in female data.

Conversely, models trained on male-only data showed balanced performance but with slightly lower effectiveness in detecting conditions in females, compared to their male counterparts. Training on the dataset with both populations yielded improvements in generalization, as seen with the nearly equivalent accuracy and sensitivity values for both male and female test data.

Table 3 offers an in-depth look at accuracy across sub-populations within the test sets, delineating performance based on the presence of recess distension (RD) and by sex. Notably, models trained on males demonstrated better accuracy in detecting no joint recess distension in females (89.78%) than in detecting the presence of RD in males (84.26%). In contrast, training on female data resulted in the highest accuracy for detecting RD in males (89.61%) but struggled with males without RD (76.92%), highlighting a variability that may stem from the different physical presentations of RD between sexes.

**Table 1. Sex-biased model performance on a sex-balanced dataset.**

|  | Accuracy | Sensitivity | Specificity | AUC |
|---|---|---|---|---|
| **Trained on Males** | 83.98% | 83.11% | **87.86%** | 0.9225 |
| **Trained on Females** | 87.47% | 88.46% | 83.1% | **0.9269** |
| **Trained on Both** | **87.73%** | **88.51%** | 84.29% | 0.9209 |

This table presents accuracy, sensitivity, specificity, and AUC metrics for models trained to detect knee synovial recess distension on male-only, female-only, and combined ultrasound datasets, tested across a sex-balanced dataset.

**Table 2. Detailed breakdown of sex-biased model performance across different sex-specific test subsets.**

|  | Accuracy | | Sensitivity | | Specificity | | AUC | |
|---|---|---|---|---|---|---|---|---|
|  | M. Sp. | F. Sp. | M. Sp. | F. Sp. | M. Sp. | F. Sp. | M. Sp. | F. Sp. |
| **Trained on Males** | 84.5% | 83.3% | 84.26% | 81.75% | **85.64%** | **89.78%** | **0.8495** | 0.8576 |
| **Trained on Females** | 87.42% | **87.87%** | **89.61%** | 87.73% | 76.92% | 88.44% | 0.8327 | **0.8809** |
| **Trained on Both** | **87.51%** | **87.87%** | 89.08% | **87.83%** | 80% | 88% | 0.8454 | 0.8792 |

The table quantifies accuracy, sensitivity, specificity, and AUC for knee synovial recess distension detection in male and female patients. M: Male. F: Female. Sp: Subpopulation in the test set.

**Table 3. Model accuracy by sex and synovial recess distension in test sub-populations.**

|  | Male Sp w/ RD | Male Sp w/o RD | Female Sp w/ RD | Female Sp w/o RD |
|---|---|---|---|---|
| Trained on Males | 84.26% | **85.64%** | 81.75% | **89.78%** |
| Trained on Females | **89.61%** | 76.92% | 87.73% | 88.44% |
| Trained on Both | 89.08% | 80% | **87.83%** | 88% |

Sp: Subpopulation in the test set. w/: with. w/o: without. RD: knee synovial recess distension.

Training on both male and female data seemed to balance these discrepancies, showing relatively high accuracy across all sub-groups, particularly in females, both with and without RD (87.83% and 88.00%, respectively), and acceptable performance in males.

Fig 2 shows a visual representation of how bias influences the model's decision-making process. By generating classification heatmaps, we highlight the focus areas of the models during RD classification. Notably, models trained exclusively on one sex tend to concentrate on different areas of the MSKUS compared to their counterparts trained on the opposite sex. Furthermore, the figure illustrates that training on both male and female data leads to heatmaps that more accurately correlate with the actual areas of synovial recess distension, as confirmed by the segmentation masks.

**Stability and robustness: 5-fold validation results.** The robustness of these findings was further substantiated by the 5-fold validation results shown in Table 4. The validation outcomes closely mirrored the test results, with female-only training again showing superior performance in terms of accuracy (mean 87.65%, SD±0.04) and AUC (mean 0.9368, SD±0.04). The models trained on both sexes exhibited slightly lower, yet consistent, performance metrics, with an accuracy mean of 86.65% (SD±0.02) and an AUC mean of 0.9269 (SD±0.01). This consistency across folds underscores the reliability of the models when subjected to varying data subsets, with female-specific data consistently leading to better model performance.

Paired bootstrapping analysis revealed significant performance differences based on training data composition. The Male-Only model consistently underperformed compared to both the Female-Only and Balanced models, indicating that training exclusively on male data led to reduced diagnostic accuracy. Our statistical findings, based on 5,000 iterations, are as follows:

- Male-Only vs. Female-Only:

  Mean Difference (Male – Female): −3.47 percentage points
  Standard Deviation of Difference: 1.13 percentage points
  95% Confidence Interval: [−5.65%, −1.41%]
  p-value: 0.0030

- Male-Only vs. Balanced:

  Mean Difference (Male – Balanced): −3.48 percentage points
  Standard Deviation of Difference: 1.07 percentage points
  95% Confidence Interval: [−5.54%, −1.41%]
  p-value: 0.0020

- Female-Only vs. Balanced:

  Mean Difference (Female – Balanced): 0.01 percentage points
  Standard Deviation of Difference: 0.97 percentage points
  95% Confidence Interval: [−1.85%, 1.85%]
  p-value: 0.5180

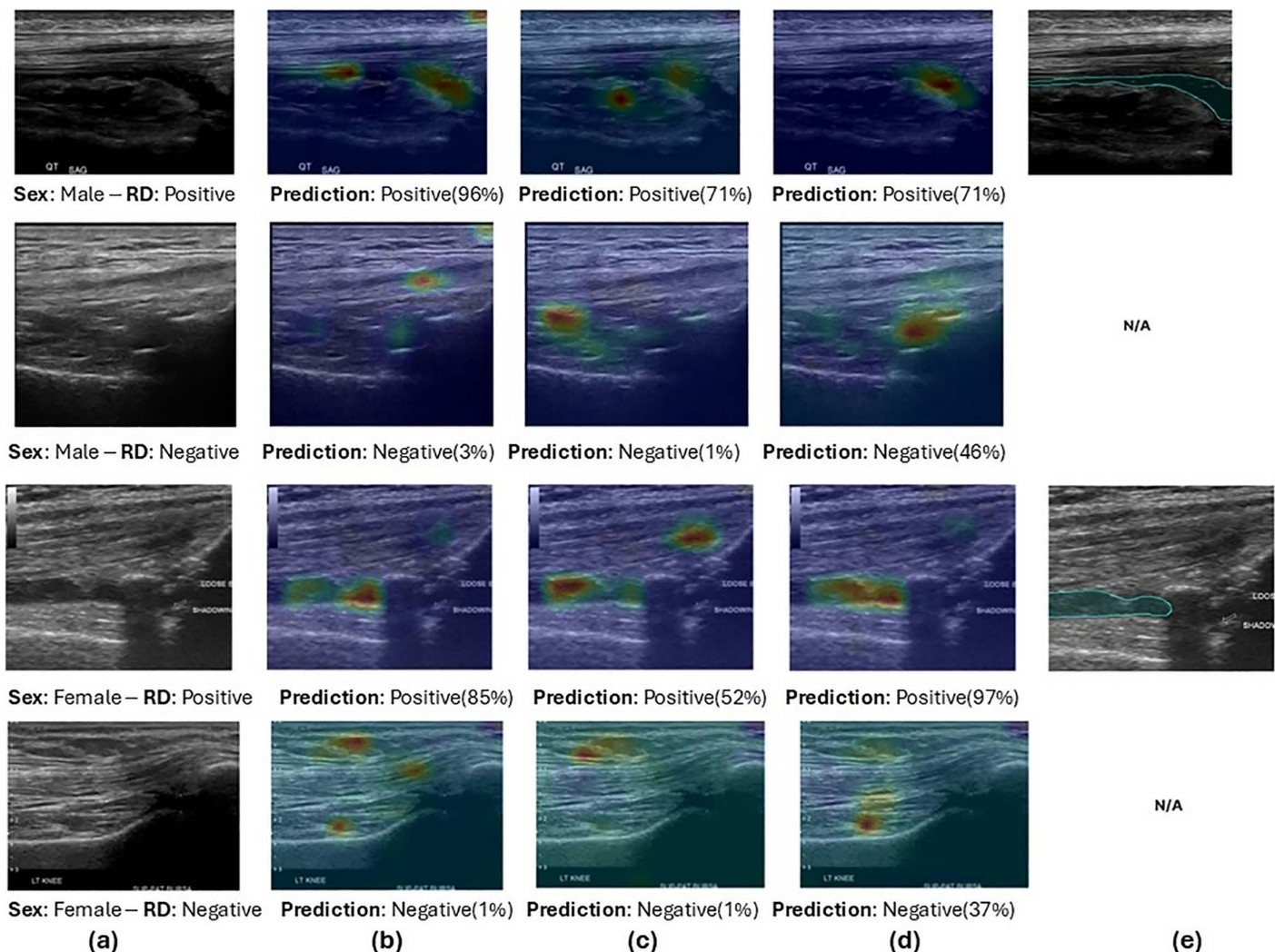

**Fig 2. Grad-CAM heatmaps showing model focus in sex-biased synovial recess distension detection.** Column (a) displays the original captured ultrasound image; column (b) shows the heatmap for models trained on male data; column (c) for models trained on female data; and column (d) for models trained on a combined dataset. Column (e) includes a segmentation mask of synovial recess distension as a reference. Model predictions are presented beneath each heatmap. Note that while model predictions align, the focus areas vary depending on the sex composition of the training data. This visualization aids in understanding how training variability impacts model precision and bias. RD: Synovial recess distension.

**Table 4. 5-fold validation results for model training with sex-specific data.**

|  | Accuracy | Sensitivity | Specificity | AUC |
|---|---|---|---|---|
| **Trained on Males** | 83.3%±0.01% | 83.69%±0.03% | 83.07%±0.03% | 89.58%±0.02% |
| **Trained on Females** | **87.65%±0.04%** | **88.95%±0.04%** | **86.54%±0.05%** | **93.68%±0.04%** |
| **Trained on Both** | 86.65%±0.02% | 87.76%±0.03% | 85.61%±0.04% | 92.69%±0.01% |

This table displays the outcomes of a 5-fold cross-validation for the machine learning knee synovial recess distension model, trained with datasets characterized by specific sex distributions (male, female, or a combination of both). The validation sets are derived directly from the respective training populations, thereby mirroring the sex distribution of each training set.

The Male-Only model showed a statistically significant reduction in accuracy compared to both the Female-Only and Balanced models (p < 0.005). In contrast, the Female-Only and Balanced models performed similarly (p = 0.5180), suggesting that female-trained models generalized as well as balanced models.

**Model for the determination of patient sex using knee joint ultrasound.** The evaluation of the BCNN classifiers for the determination of patient sex using knee joint ultrasound images further demonstrated the model's capabilities in a sex classification context. The performance of this classifier, when applied to the balanced test dataset, reflected high levels of accuracy and reliability.

In the structured testing environment, the classifier achieved an impressive accuracy of 95%, with a sensitivity of 96.8% and a specificity of 93.3%. The AUC was recorded at 0.967, indicating an excellent ability of the model to distinguish between male and female patients based solely on ultrasound images of the knee. The robustness of these results was further corroborated through 5-fold cross-validation, mirroring the consistency seen in direct testing. Across different validation folds, the classifier maintained similar levels of accuracy, sensitivity, specificity, and AUC, underscoring the model's stability and reliability across various subsets of data. The high accuracy of the sex classifier confirms that deep learning models can distinguish sex-related anatomical differences in knee ultrasound images. This finding underscores the influence of patient sex on learned feature representations in AI-driven musculoskeletal diagnosis. If diagnostic models inadvertently rely on these sex-specific features, they may introduce bias, particularly when trained on unbalanced datasets. Understanding these inherent differences is therefore crucial for developing fair and reliable AI-based medical imaging applications.

To further interpret the decision-making process of the sex-classification model, we utilized Grad-CAM to visualize the regions of interest in ultrasound images that contributed most significantly to the model's predictions. Fig 3 illustrates heatmaps generated for both male and female classifications. The model primarily focuses on tissue structures surrounding the knee joint, suggesting that differences in soft tissue characteristics or imaging artifacts might be leveraged for classification. However, some heatmaps reveal a reliance on non-anatomical regions, which raises potential concerns regarding bias in the learned representations. These findings highlight the importance of interpretability methods in medical AI to ensure that models base their decisions on clinically relevant features rather than spurious correlations.

## Discussion

Our study highlights the presence of sex bias in AI-driven musculoskeletal ultrasound diagnosis, revealing that models trained on male-only data perform worse than those trained on female-only or balanced datasets. These findings emphasize the need for careful dataset curation to mitigate performance disparities and improve clinical applicability. Given the widespread integration of AI in medical imaging, it is critical to address demographic biases that could disproportionately impact certain patient groups. The exploration of binary convolutional neural network (BCNN) classifiers in this study provides important insights into the performance of AI models in medical imaging, specifically in the context of detecting synovial recess distension and determining patient sex from knee joint ultrasound images. The results highlight several key aspects of AI deployment in clinical settings, particularly the influence of inherent biases and the importance of comprehensive model training.

Our findings indicate a notable variation in model performance based on the sex of the data on which the models were trained (Table 2). Specifically, models trained exclusively on female data exhibited higher accuracy and sensitivity in detecting synovial recess distension compared to those trained solely on male data. However, these same models demonstrated a drop in specificity when applied to male test images, suggesting a potential overfitting to characteristics more prevalent in female data. This is a significant concern as it highlights how even seemingly neutral variables like sex can influence an AI model's diagnostic capabilities, leading to skewed or biased outcomes.

The paired bootstrapping results confirm that demographic biases in training data significantly affect model performance. Our analysis demonstrates that models trained solely on male data perform consistently worse than those trained

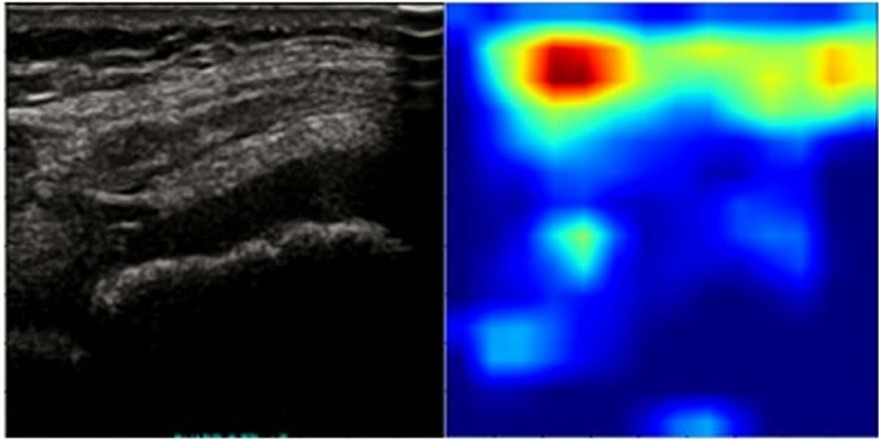

**Sex: Male**          **Sex Prediction: Male (100%)**

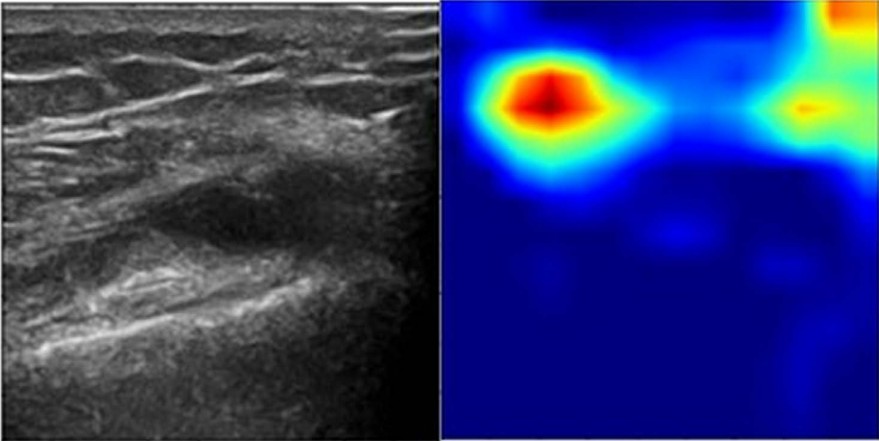

**Sex: Female**          **Sex Prediction: Female (100%)**

**Fig 3. Grad-CAM visualizations of the sex-classification model applied to knee joint ultrasound images.** The heatmaps highlight the regions that contribute most to the model's predictions, illustrating potential sex-specific imaging features. These visualizations provide insights into the model's decision-making process and help assess possible sources of bias in classification.

on female-only or balanced datasets. This suggests that male ultrasound images alone may not provide sufficient variability for robust generalization, highlighting the need for diverse datasets to achieve equitable AI performance in musculoskeletal ultrasound.

Moreover, the significant performance gap between Male-Only and Female-Only models (p = 0.0030) suggests that sex differences in musculoskeletal anatomy influence AI-based classification. Given the lack of significant performance differences between the Female-Only and Balanced models (p = 0.5180), these findings further indicate that models trained on female-only data generalize more effectively.

These results reinforce the necessity of balanced, demographically diverse training datasets to ensure fair and reliable diagnostic performance across patient populations. Future studies should investigate whether additional factors, such

as age and anatomical variations, further impact model performance, as well as explore strategies like domain adaptation or augmentation techniques to mitigate dataset imbalances. Our results align with previous studies demonstrating demographic biases in AI models trained on imbalanced datasets. For example, Banerjee et al. [30] and Larrazabal et al. [31] found that sex-based differences in training data significantly influenced the sensitivity and specificity of diagnostic AI models. Similarly, Tripathi et al. [32] highlighted that models trained on a single demographic subgroup tend to overfit to their characteristics, reducing generalizability. Our findings extend this work by providing robust statistical evidence through paired bootstrapping analysis, confirming that male-only training yields lower diagnostic performance compared to female-only or balanced training scenarios.

The use of classification heatmaps further revealed that models trained on data from one sex focused on different anatomical areas compared to models trained on the opposite sex (Fig 2). This discrepancy underscores a form of sex bias, where the AI's decision-making process varies not by the pathology present but by the demographic characteristics of the training data. Such findings are critical for clinical applications, as they suggest that deploying these models without adequate cross-demographic validation could lead to differential diagnostic outcomes, undermining the equity and efficacy of healthcare services.

Bias mitigation is an active area of research [33,34]. One effective approach is balancing the training dataset to ensure it equally represents all demographic groups involved in the study. This helps prevent the model from overfitting to features that are overrepresented in an unbalanced dataset. Additionally, techniques such as augmentation of underrepresented groups in the training data, cross-validation across different demographic groups, and the use of ensemble methods that integrate multiple models trained on diverse subsets can further enhance the fairness and robustness of AI systems.

The presence of sex-based bias as demonstrated in our results aligns with broader literature that acknowledges anatomical differences across various demographics in MSK ultrasound imaging [18]. However, our study extends this by quantitatively assessing the impact of these differences on AI model performance. By highlighting the specific ways in which biases can manifest, our study advocates for a more nuanced approach to training AI systems in healthcare—suggesting that diversity in training data is not just beneficial but essential for developing robust medical AI.

## Limitations and prospects for future research

A potential limitation of our study is that the preliminary image review was conducted by a single expert sonographer with extensive experience (32 years). While the expert followed a strict protocol to ensure consistency in image selection, the involvement of only one reviewer may introduce a degree of subjectivity. In future studies, a multi-expert validation approach will be implemented to assess inter-rater reliability and mitigate potential operator bias.

While the initial recess distension detection accuracy of approximately 85% may be considered below optimal clinical standards, our analysis underscores the importance of dataset size and diversity in improving AI performance.

One significant limitation of our study is the potential impact of inter-operator variability, which is a well-known challenge in ultrasound imaging. Ultrasound is operator-dependent, and variations in how different technicians capture images can introduce inconsistencies that may affect the training and performance of AI models. Addressing this limitation would require standardizing image acquisition procedures or developing algorithms robust enough to handle variations inherent in operator-dependent data.

Further, the lack of additional demographic and technical data in our dataset limits our ability to explore other potential biases that could affect diagnostic accuracy, such as age-specific or ethnically-inherent anatomical differences, or even variations attributable to different ultrasound equipment manufacturers. Future research should aim to include a broader range of demographic and technical factors to fully understand the complexities of bias in medical AI in the application of ultrasound diagnosis.

Bias in AI models is primarily driven by dataset composition rather than architecture choice. Regardless of whether CNNs or ViTs are used, training data diversity and representativeness remain the most critical factors in ensuring fairness and mitigating bias in diagnostic models. Future research could investigate whether ViTs amplify or mitigate demographic biases in musculoskeletal ultrasound classification by leveraging their attention-based feature extraction.

Self-supervised learning (SSL) methods, such as DINOv2 and SimCLR, have demonstrated strong generalization capabilities in medical imaging, particularly in scenarios where labeled data is limited. These approaches leverage large-scale, unlabeled datasets for pretraining, potentially enabling models to learn robust feature representations that generalize across demographic groups. Incorporating SSL-based pretraining could influence how models learn from imbalanced demographic distributions and mitigate bias in musculoskeletal ultrasound classification. Future work should evaluate SSL-based architectures in this context, comparing their ability to reduce bias against conventional supervised learning approaches.

The rapid advancement of foundation models, such as SAM and MedCLIP, has introduced new opportunities for improving generalization and fairness in medical imaging. These models, pretrained on large-scale multimodal datasets, have demonstrated robust performance in zero-shot and fine-tuned settings across various medical tasks. However, their effectiveness in mitigating demographic bias in musculoskeletal ultrasound classification remains largely unexplored. Future research should investigate how foundation models perform in terms of sex bias, both in zero-shot inference and when fine-tuned on domain-specific ultrasound data. While our current study focuses on CNN-based classifiers due to their clinical relevance and computational feasibility, extending this analysis to foundation models could provide further insights into mitigating bias in AI-driven diagnostics.

## Conclusions

In conclusion, our study underscores the critical need for addressing demographic biases in AI-driven MSKUS diagnostics for synovial recess distension. By leveraging binary convolutional neural network classifiers, we showed variability in AI performance linked to demographic features such as sex, which could potentially skew clinical outcomes if unaddressed. Despite strong classification metrics, our findings highlight that these models may not consistently perform across diverse clinical scenarios. The study reiterates the importance of incorporating robust, diverse datasets and cross-demographic validation to mitigate bias and enhance diagnostic accuracy. Looking forward, refining AI models to handle variations inherent in ultrasound imaging, and broadening the demographic scope of datasets are essential steps to support the reliable application of AI in clinical settings. Future work should assess whether these limitations persist when using SSL approaches, foundation models, and ViTs.

## Supporting information

**S1 Fig. Overview of the study workflow for AI-driven musculoskeletal ultrasound analysis.** The pipeline consists of three main stages: (1) Data Preparation, including expert screening, dataset composition, and partitioning; (2) Model Training, utilizing EfficientNet-B4 with different training scenarios and data augmentation strategies; and (3) Evaluation and Bias Analysis, which involves performance assessment, Grad-CAM visualization, and bias comparison across training conditions.
(TIF)

**S2 Fig. Grad-CAM heatmaps showing model focus in sex-biased synovial recess distension detection.** Column (a) displays the original captured ultrasound image; column (b) shows the heatmap for models trained on male data; column (c) for models trained on female data; and column (d) for models trained on a combined dataset. Column (e) includes a segmentation mask of synovial recess distension as a reference. Model predictions are presented beneath each heatmap. Note that while model predictions align, the focus areas vary depending on the sex composition of the training data.

This visualization aids in understanding how training variability impacts model precision and bias. RD: Synovial recess distension.
(TIF)

**S3 Fig. Grad-CAM visualizations of the sex-classification model applied to knee joint ultrasound images.** The heatmaps highlight the regions that contribute most to the model's predictions, illustrating potential sex-specific imaging features. These visualizations provide insights into the model's decision-making process and help assess possible sources of bias in classification.
(TIF)

**S1 Table. Sex-biased model performance on a sex-balanced dataset.** This table presents accuracy, sensitivity, specificity, and AUC metrics for models trained to detect knee synovial recess distension on male-only, female-only, and combined ultrasound datasets, tested across a sex-balanced dataset.
(DOCX)

**S2 Table. Detailed breakdown of sex-biased model performance across different sex-specific test subsets.** The table quantifies accuracy, sensitivity, specificity, and AUC for knee synovial recess distension detection in male and female patients. M: Male. F: Female. Sp: Subpopulation in the test set.
(DOCX)

**S3 Table. Model accuracy by sex and synovial recess distension in test sub-populations.** Sp: Subpopulation in the test set. w/: with. w/o: without. RD: knee synovial recess distension.
(DOCX)

**S4 Table. 5-fold validation results for model training with sex-specific data.** This table displays the outcomes of a 5-fold cross-validation for the machine learning knee synovial recess distension model, trained with datasets characterized by specific sex distributions (male, female, or a combination of both). The validation sets are derived directly from the respective training populations, thereby mirroring the sex distribution of each training set.
(DOCX)

## Author contributions

**Conceptualization:** M. Mendez, Pascal N Tyrrell.

**Data curation:** M. Mendez, S. Demello, C. Lee, M. Dang.

**Formal analysis:** M. Mendez, Pascal N Tyrrell.

**Funding acquisition:** Pascal N Tyrrell.

**Investigation:** M. Mendez.

**Methodology:** M. Mendez, N. Jafarpisheh, S. Demello, C. Lee, M. Dang.

**Project administration:** Pascal N Tyrrell.

**Software:** N. Jafarpisheh.

**Supervision:** M. Mendez.

**Validation:** M. Mendez, N. Jafarpisheh, S. Demello, C. Lee, M. Dang, Pascal N Tyrrell.

**Visualization:** M. Mendez.

**Writing – original draft:** M. Mendez, Pascal N Tyrrell.

**Writing – review & editing:** N. Jafarpisheh, S. Demello, C. Lee, M. Dang.

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
