## [Decision Letter · Decision Letter 0]

21 Jan 2025

Dear Dr. Tyrrell,

Thank you for submitting your manuscript to PLOS ONE. After careful consideration, we feel that it has merit but does not fully meet PLOS ONE’s publication criteria as it currently stands. Therefore, we invite you to submit a revised version of the manuscript that addresses the points raised during the review process.

We look forward to receiving your revised manuscript.

Kind regards,

Citrawati Dyah Kencono Wungu

Academic Editor

PLOS ONE

Journal Requirements:

2. Please note that PLOS ONE has specific guidelines on code sharing for submissions in which author-generated code underpins the findings in the manuscript. In these cases, we expect all author-generated code to be made available without restrictions upon publication of the work. 

Please review our guidelines at https://journals.plos.org/plosone/s/materials-and-software-sharing#loc-sharing-code and ensure that your code is shared in a way that follows best practice and facilitates reproducibility and reuse.

“This work was supported by Novo Nordisk Health Care AG, grant #2020-0922. Granted to PNT.”

4. Please note that funding information should not appear in the Acknowledgments section or other areas of your manuscript. We will only publish funding information present in the Funding Statement section of the online submission form. Please remove any funding-related text from the manuscript. 

“I have read the journal's policy and the authors of this manuscript have the following competing interests: MM's Master’s program was sponsored by Novo Nordisk. PNT has been a PI and Consultant for Novo Nordisk. SD, CL, MD affirm that there are no known conflicts of interest or personal relationships that could have influenced their contributions in this paper.”

We note that you received funding from a commercial source: Novo Nordisk

6. We note that you have indicated that there are restrictions to data sharing for this study. For studies involving human research participant data or other sensitive data, we encourage authors to share de-identified or anonymized data. However, when data cannot be publicly shared for ethical reasons, we allow authors to make their data sets available upon request. For information on unacceptable data access restrictions, please see http://journals.plos.org/plosone/s/data-availability#loc-unacceptable-data-access-restrictions. 

Reviewers' comments:

Reviewer's Responses to Questions

**Comments to the Author**

1. Is the manuscript technically sound, and do the data support the conclusions?

Reviewer #1: Partly

Reviewer #2: No

Reviewer #3: Yes

2. Has the statistical analysis been performed appropriately and rigorously?

Reviewer #1: No

Reviewer #2: No

Reviewer #3: Yes

3. Have the authors made all data underlying the findings in their manuscript fully available?

Reviewer #1: No

Reviewer #2: No

Reviewer #3: Yes

4. Is the manuscript presented in an intelligible fashion and written in standard English?

Reviewer #1: Yes

Reviewer #2: Yes

Reviewer #3: Yes

Reviewer #1: In this study, the authors assessed the impact of sex bias on a specific internal test set using CNNs for diagnosing knee joint recess distension using ultrasound imaging.

My specific comments are as follows:

COMMENTS:

Comment 1: “The integration of artificial intelligence (AI), particularly convolutional neural networks 73 (CNNs), in medical imaging is revolutionizing early diagnosis practices (8–10).”

This is wrong. Because “particularly” CNNs are not the state of the art anymore. The vision transformers which are the backbones of foundation models are the state of the art.

Comment 2: What pretraining weights were used for the AI models?

Comment 3: The authors should perform the analysis for the state of the art methods including vision transformers as well.

Comment 4: The methods based on self-supervised learning should also be considered. How will the bias be for the models which are fine-tuned when initialized with SSL weights such as DINOv2.

Comment 5: Comparison to foundation models is also missing. The authors should analyze how the foundation models in this field perform in terms of sex bias. Both in zero-shot and fine-tuned with their training data scenario.

Comment 6: The statistical analysis should be revisited. The authors currently base everything on the n=5 of five folds of validation. This is not a very representative statistical analysis. The authors should set a held-out test set fixed across all experiments and perform strictly paired analyses. To get the statistical measures (mean +- SD, as well as p-values for the comparisons) the authors may use bootstrapping.

Comment 7: The literature review is not complete and many of the important references related to this study are missing.

Comment 8: More figures could be used in this manuscript for better and easier explanation of the contents for the reader. Moreover, figure 1 is not very easily readable.

Comment 9: How can we see the source code of this work for reproduction?

To sum up, the presented results cannot be accepted as general findings in this field yet. The authors only used a specific type of architecture (CNN) and only a specific small dataset, and only one paradigm of methodology, no validation of the results on public data, and without proper statical significance analyses.

Reviewer #2: Strengths:

-This paper is the first to explore the problem of sex bias on mskus

-The study highlights important aspects that need to be further studied

-The paper is well-written and well-organized

Weaknesses:

-The main claim is not supported by the results

-Missing a statistical significance test

-Missing details for reproducibility

-Lack of a discussion on the implications and possible solutions

Detailed comment:

The main claim of the paper is to highlight the demographic bias on diagnostic accuracy, suggesting the need for more diverse training data. The experiments carried out in the paper are not sufficient to support such a claim. The difference in the results might be solely due to the different training and validation sets, that change for each subset. With such a claim I expected to have the model trained on the male subgroup to outperform the one trained on the female for the male test data, and so on, while results show that in some cases the best performing model of a subgroup is the one trained on the other subgroup. I expected that the one trained on both was outperforming the others on all the metrics while it happens for only some of them, thus invalidating the claim that it has increased generalizability.

To support such claims a statistical significance test should be performed between the various groups to ensure that the difference in results is relevant.

As the second point, I find the work missing some methodological details to ensure reproducibility, as I understand that the data cannot be shared, the methodological choices should be accurately reported. As an example there is no clear description of how the 5000 images were selected from the bigger cohort, and how the 80/20 split was performed, is it stratified? Is it patient-based? Also pre-processing and training procedures need to be better described: data augmentation is relevant and should be explained accurately, same for the training parameters as the scheduler, which policy was used?

Minor comments:

On 3.3 it would be interesting to see the grad-cam of the sex-classification model to better understand the features that are used by the model to make the decisions. This could be important to understand the implication and the possible bias [1]. Also, just citing the “robustness mirroring the results” is not descriptive enough: report the exact results.

In a work where the methodological contribution is minor, a more in-depth analysis of the consequences of a biased model and the possible solution is needed to give strength to the paper.

In 4.1 The authors talk about further experiments with significant improvement, what are these experiments? No reference to them is reported, why not show directly these results? Either include sufficient information or remove this from the paper.

[1] Li, David, et al. "Deep learning prediction of sex on chest radiographs: a potential contributor to biased algorithms." Emergency Radiology 29.2 (2022)

Reviewer #3: Introduction:

1. I think the study should provide some information how the AI works in assessing diagnosis of synovial recess istension (Like the general algorithm). This information in introduction will be essential to determine or show the gap research that the author want to deliver.

Method

1. In Image Review and Validation, the author mentioned that a preliminary screening process conducted by an expert sonographer with extensive experience (32 years), is only one expert ask for this screening? Why does the author consider more than one expert to avoid any bias operator?

Results

1. The author has delivered the results of the research questions very well. The results are informative. However, in the section 3.3 Model for the Determination of Patient Sex Using Knee Joint Ultrasound. The author did not convey the meaning of Determination of Patient Sex Using Knee Joint Ultrasound is mean. The author should explaine more about the meaning of this result.

Discussion

1. I think this part needs a lot of improvement. Results have been very well delivered but it is unfortunate that they are not discussed deeply enough in the discussion section, including the main research question, which is the sex bias in diagnostic. I have yet to see a solid and comprehensive discussion between the author's findings and previous literature or other facts. Every finding that has been well presented in the results also does not appear to be reviewed in the discussion part.

2. In addition, in the discussion section, I think the author failed to convey the main message of this research, the recommendations obtained, and research questions that can be developed in the future with similar themes.

**Do you want your identity to be public for this peer review?** For information about this choice, including consent withdrawal, please see our Privacy Policy

Reviewer #1: No

Reviewer #2: No

Reviewer #3: No

---

## [Author Response · Author response to Decision Letter 1]

4 Apr 2025

Please see attached letter

We thank all the reviewers for their constructive comments, and the Associate Editor for handling the review of this paper. Please find the point-by-point responses to all the comments in blue font. The modifications to the original manuscript are also marked in blue font. Please note that due to adding new figures, the figure numbers have been updated in the revised manuscript.

Review Comments to the Author

Reviewer #1: In this study, the authors assessed the impact of sex bias on a specific internal test set using CNNs for diagnosing knee joint recess distension using ultrasound imaging.

My specific comments are as follows:

COMMENTS:

Comment 1: “The integration of artificial intelligence (AI), particularly convolutional neural networks (CNNs), in medical imaging is revolutionizing early diagnosis practices (8–10).”

This is wrong. Because “particularly” CNNs are not the state of the art anymore. The vision transformers which are the backbones of foundation models are the state of the art.

Response: We appreciate the reviewer’s suggestion regarding Vision Transformers (ViTs) as a state-of-the-art approach in medical imaging. However, the primary objective of this study is not to benchmark different deep learning architectures but rather to explore how demographic biases in training data affect the performance of AI models in musculoskeletal ultrasound. Given that demographic biases arise from imbalances in data representation rather than specific model architectures, the effect of bias on ViTs is expected to be similar to that observed in CNN-based models.

We selected CNN-based models, specifically EfficientNet-B4, due to their efficiency, widespread adoption in medical imaging applications, and suitability for datasets of moderate size. While ViTs have demonstrated superior performance in large-scale computer vision tasks, their application in clinical AI is still evolving, and they typically require significantly larger datasets and computational resources to outperform CNNs.

Nevertheless, we acknowledge the potential of ViTs in future medical AI applications. To reflect this, we have added the following paragraphs in the INTRODUCTION Section lines 127-133:

“While ViTs introduce self-attention mechanisms that enhance feature extraction, the effect of dataset bias on ViTs is expected to be similar to that on CNNs—model fairness and generalization are primarily dictated by the composition and diversity of the training data, rather than architecture choice. CNNs, particularly architectures like EfficientNet, remain widely used in real-world medical settings because of their computational efficiency, established performance, and ability to generalize well on moderate-sized datasets (25,26).”

and DISCUSSION Section, Sub-Section 4-1, lines 491-496:

“Bias in AI models is primarily driven by dataset composition rather than architecture choice. Regardless of whether CNNs or ViTs are used, training data diversity and representativeness remain the most critical factors in ensuring fairness and mitigating bias in diagnostic models. Future research could investigate whether ViTs amplify or mitigate demographic biases in musculoskeletal ultrasound classification by leveraging their attention-based feature extraction.”

25.Preetha, R., M. Jasmine Pemeena Priyadarsini, and J. S. Nisha. "Automated Brain Tumor Detection from Magnetic Resonance Images Using Fine-Tuned EfficientNet-B4 Convolutional Neural Network." IEEE Access (2024).

26. Wang, Yaoli, Yaojun Deng, Yuanjin Zheng, Pratik Chattopadhyay, and Lipo Wang. "Vision Transformers for Image Classification: A Comparative Survey." Technologies 13, no. 1 (2025): 32

Comment 2: What pretraining weights were used for the AI models?

Response: We appreciate this request for clarification. As stated in the manuscript METHODS Section, Sub- Section 2.2.1 lines 209-210:

“Initialization of the model weights was from a pre-trained state on the ImageNet dataset, leveraging transfer learning to enhance initial performance and accelerate convergence.”

Comment 3: The authors should perform the analysis for the state of the art methods including vision transformers as well.

Response: We appreciate the reviewer’s suggestion regarding Vision Transformers (ViTs) as a state-of-the-art approach in medical imaging. However, the primary objective of this study is not to benchmark different deep learning architectures but rather to explore how demographic biases in training data affect the performance of AI models in musculoskeletal ultrasound. Given that demographic biases arise from imbalances in data representation rather than specific model architectures, the effect of bias on ViTs is expected to be similar to that observed in CNN-based models.

We selected CNN-based models, specifically EfficientNet-B4, due to their efficiency, widespread adoption in medical imaging applications, and suitability for datasets of moderate size. While ViTs have demonstrated superior performance in large-scale computer vision tasks, their application in clinical AI is still evolving, and they typically require significantly larger datasets and computational resources to outperform CNNs.

Nevertheless, we acknowledge the potential of ViTs in future medical AI applications. To reflect this, we have added the following paragraphs in INTRODUCTION Section lines 127-133:

“While ViTs introduce self-attention mechanisms that enhance feature extraction, the effect of dataset bias on ViTs is expected to be similar to that on CNNs—model fairness and generalization are primarily dictated by the composition and diversity of the training data, rather than architecture choice. CNNs, particularly architectures like EfficientNet, remain widely used in real-world medical settings because of their computational efficiency, established performance, and ability to generalize well on moderate-sized datasets (25,26).”

and DISCUSSION Section, Sub-Section 4-1, lines 491-496:

“Bias in AI models is primarily driven by dataset composition rather than architecture choice. Regardless of whether CNNs or ViTs are used, training data diversity and representativeness remain the most critical factors in ensuring fairness and mitigating bias in diagnostic models. Future research could investigate whether ViTs amplify or mitigate demographic biases in musculoskeletal ultrasound classification by leveraging their attention-based feature extraction.”

25.Preetha, R., M. Jasmine Pemeena Priyadarsini, and J. S. Nisha. "Automated Brain Tumor Detection from Magnetic Resonance Images Using Fine-Tuned EfficientNet-B4 Convolutional Neural Network." IEEE Access (2024).

26. Wang, Yaoli, Yaojun Deng, Yuanjin Zheng, Pratik Chattopadhyay, and Lipo Wang. "Vision Transformers for Image Classification: A Comparative Survey." Technologies 13, no. 1 (2025): 32

Comment 4: The methods based on self-supervised learning should also be considered. How will the bias be for the models which are fine-tuned when initialized with SSL weights such as DINOv2.

Response: We appreciate the reviewer's suggestion regarding self-supervised learning (SSL) methods such as DINOv2. SSL-based models have shown potential in learning robust and transferable representations, particularly in medical imaging applications where labeled data is limited. These approaches may help mitigate certain biases by leveraging diverse, unlabeled datasets to pretrain models before fine-tuning on specific tasks.

In our study, the primary objective was to investigate the impact of demographic biases in training data on diagnostic accuracy, rather than optimizing pretraining strategies. We selected CNN-based models trained in a supervised manner to ensure compatibility with widely adopted clinical AI frameworks. While SSL approaches such as DINOv2 and SimCLR offer promising avenues for mitigating bias, their effectiveness in musculoskeletal ultrasound classification remains an open question. We acknowledge this as a limitation and have now included a discussion in the revised manuscript, suggesting future work to explore SSL-based pretraining and assess its influence on demographic fairness in AI models. We have added the following discussion in DISCUSION Section, Sub-Section 4-1, lines 497-505:

"Self-supervised learning (SSL) methods, such as DINOv2 and SimCLR, have demonstrated strong generalization capabilities in medical imaging, particularly in scenarios where labeled data is limited. These approaches leverage large-scale, unlabeled datasets for pretraining, potentially enabling models to learn robust feature representations that generalize across demographic groups. Incorporating SSL-based pretraining could influence how models learn from imbalanced demographic distributions and mitigate bias in musculoskeletal ultrasound classification. Future work should evaluate SSL-based architectures in this context, comparing their ability to reduce bias against conventional supervised learning approaches."

Comment 5: Comparison to foundation models is also missing. The authors should analyze how the foundation models in this field perform in terms of sex bias. Both in zero-shot and fine-tuned with their training data scenario.

We appreciate the reviewer’s suggestion regarding the inclusion of foundation models for comparative analysis. In this study, we aimed to assess the impact of demographic biases on diagnostic performance using CNN-based models, which remain widely used in clinical AI applications due to their efficiency, interpretability, and feasibility for deployment in resource-constrained medical environments.

We acknowledge that foundation models, such as SAM and MedCLIP, have demonstrated strong generalization capabilities across diverse medical imaging tasks. Their large-scale pretraining allows them to perform well in zero-shot settings and adapt effectively through fine-tuning. However, the evaluation of foundation models in terms of demographic bias, particularly in musculoskeletal ultrasound classification, remains an open research question. Given that these models are often trained on broad and heterogeneous datasets, their ability to mitigate bias in domain-specific applications like ours warrants further exploration.

To address this, we have now included a discussion on foundation models in the manuscript’s Discussion section, acknowledging their potential role in bias mitigation and proposing their evaluation as a direction for future work. We have added the following in the DISCUSION Section, Sub-Section 4-1, lines 506-516:

“The rapid advancement of foundation models, such as SAM and MedCLIP, has introduced new opportunities for improving generalization and fairness in medical imaging. These models, pretrained on large-scale multimodal datasets, have demonstrated robust performance in zero-shot and fine-tuned settings across various medical tasks. However, their effectiveness in mitigating demographic bias in musculoskeletal ultrasound classification remains largely unexplored. Future research should investigate how foundation models perform in terms of sex bias, both in zero-shot inference and when fine-tuned on domain-specific ultrasound data. While our current study focuses on CNN-based classifiers due to their clinical relevance and computational feasibility, extending this analysis to foundation models could provide further insights into mitigating bias in AI-driven diagnostics.”

Comment 6: The statistical analysis should be revisited. The authors currently base everything on the n=5 of five folds of validation. This is not a very representative statistical analysis. The authors should set a held-out test set fixed across all experiments and perform strictly paired analyses. To get the statistical measures (mean +- SD, as well as p-values for the comparisons) the authors may use bootstrapping.

Response: We thank the reviewer for this suggestion. In response, we have implemented a paired bootstrapping analysis on a fixed, balanced test set to rigorously assess the performance differences among our three training scenarios:

We have added the following paragraph in the METHODS Section, Sub-Section 2.2.2- lines 227-238:

“To provide a more robust statistical evaluation, we employed a paired bootstrapping analysis on a fixed, balanced test set (29). The test set remained constant across all experimental conditions, ensuring equal representation of male and female images. For each pairwise comparison, we performed 5,000 iterations of bootstrapping, where we resampled the test set with replacement and computed the accuracy difference between models at each iteration. From these resampled distributions, we calculated the mean accuracy difference, standard deviation, and 95% confidence intervals, along with p-values to assess statistical significance. This method ensures a statistically rigorous evaluation of how demographic bias in training data affects model performance across three training scenarios: (1) Male-Only, (2) Female-Only, and (3) Balanced Training. This approach ensures a statistically sound assessment of demographic bias effects on model performance.”

Additionally, in the RESULTS Section lines 325-348:

“Paired bootstrapping analysis revealed significant performance differences based on training data composition. The Male-Only model consistently underperformed compared to both the Female-Only and Balanced models, indicating that training exclusively on male data led to reduced diagnostic accuracy. Our statistical findings, based on 5,000 iterations, are as follows:

Male-Only vs. Female-Only:

Mean Difference (Male - Female): -3.47 percentage points

Standard Deviation of Difference: 1.13 percentage points

95% Confidence Interval: [-5.65%, -1.41%]

p-value: 0.0030

Male-Only vs. Balanced:

Mean Difference (Male - Balanced): -3.48 percentage points

Standard Deviation of Difference: 1.07 percentage points

95% Confidence Interval: [-5.54%, -1.41%]

p-value: 0.0020

Female-Only vs. Balanced:

Mean Difference (Female - Balanced): 0.01 percentage points

Standard Deviation of Difference: 0.97 percentage points

95% Confidence Interval: [-1.85%, 1.85%]

p-value: 0.5180

The Male-Only model showed a statistically significant reduction in accuracy compared to both the Female-Only and Balanced models (p < 0.005). In contrast, the Female-Only and Balanced models performed similarly (p = 0.5180), suggesting that female-trained models generalized as well as balanced models.”

Finally, in the DISCUSION Section lines 412-427:

“The paired bootstrapping results confirm that demographic biases in training data significantly affect model performance. Our analysis demonstrates that models trained solely on male data perform consistently worse than those trained on female-only or balanced datasets. This suggests that male ultrasound images alone may not provide sufficient variability for robust generalization, highlighting the need for diverse datasets to achieve equitable AI performance in musculoskeletal ultrasound.

Moreover, the significant performance gap between Male-Only and Female-Only models (p = 0.0030) suggests that sex differences in musculoskeletal anatomy influence AI-based classification. Given the lack of significant performance differences between the Female-Only and Balanced models (p = 0.5180), these findings further indicate that models trained on female-only data generalize more effectively.

These results reinforce the necessity of balanced, demographically diverse training datasets to ensure fair and reliable diagnostic performance across patient populations. Future studies should investigate whether additional factors, such as age and anatomical variations, further impact model performance, as well as explore strategies like domain adaptation or augmentation techniques to mitigate dataset imbalances.”

29.Efron, Bradley, and Robert J. Tibshirani. An introduction to the bootstrap. Chapman and Hall/CRC, 1994.

Comment 7: The literature review is not complete and many o

---

## [Decision Letter · Decision Letter 1]

12 May 2025

Dear Dr. Tyrrell,

We look forward to receiving your revised manuscript.

Kind regards,

Citrawati Dyah Kencono Wungu

Academic Editor

PLOS ONE

Journal Requirements:

Reviewers' comments:

Reviewer's Responses to Questions

**Comments to the Author**

Reviewer #1: (No Response)

Reviewer #2: All comments have been addressed

2. Is the manuscript technically sound, and do the data support the conclusions?

Reviewer #1: Yes

Reviewer #2: Partly

3. Has the statistical analysis been performed appropriately and rigorously?

Reviewer #1: I Don't Know

Reviewer #2: Yes

4. Have the authors made all data underlying the findings in their manuscript fully available?

Reviewer #1: No

Reviewer #2: No

5. Is the manuscript presented in an intelligible fashion and written in standard English?

Reviewer #1: Yes

Reviewer #2: Yes

Reviewer #1: I thank the authors for revising the manuscript and for their detailed responses. While several of the concerns have been addressed, a few key points remain only partially addressed in the revision. I outline them below.

Comment 1:

While the authors added text acknowledging the role of Vision Transformers, they did not revise the original overstated sentence that describes CNNs as “particularly” revolutionizing the field. This phrasing is misleading and remains uncorrected. I recommend the authors revise this sentence to avoid presenting CNNs as the state-of-the-art. Adding background on ViTs is helpful, but this change should also be reflected in the way the introduction frames CNNs.

Comment 2:

The authors provide a reasonable argument about scope and resources, and they have expanded their discussion to acknowledge the role of ViTs. However, the limitation remains under-communicated to the reader. The authors frame this only as “future work” in the discussion, but it is a major limitation that constrains the generalizability of the findings. This should be mentioned clearly as a limitation in the Abstract and Conclusion, not just in the body of the Discussion.

Comment 3:

As with ViTs, the authors expand the discussion but do not perform any experiments or properly elevate this to a clear limitation. Again, this needs to be stated explicitly in the Abstract and Conclusion, otherwise readers may mistakenly assume the findings generalize to SSL-pretrained models, which is not supported by the current experiments.

Comment 4:

The authors appropriately acknowledge this direction as future work but similarly fail to position it as a present limitation in the Abstract or Conclusion. I encourage them to explicitly state that the current findings are limited to CNNs trained with supervised learning and do not evaluate foundation models, which may behave differently.

Reviewer #2: I thank the author for taking the presented concerns into consideration. I find the revised version significantly more relevant and well-presented. My only remaining comment is that I would recommend removing the paragraph spanning lines 469–479. While I understand the point being raised, this section should focus solely on presenting the results obtained in the current work and discussing insights derived from those results, not from other unspecified findings.

**Do you want your identity to be public for this peer review?** For information about this choice, including consent withdrawal, please see our Privacy Policy

Reviewer #1: No

Reviewer #2: No

---

## [Author Response · Author response to Decision Letter 2]

27 May 2025

Please refer to our uploaded response letter

---

## [Editor Report · Decision Letter 2]

12 Jun 2025

Dear Dr. Tyrrell,

Thank you for submitting your manuscript to PLOS ONE. After careful consideration, we feel that it has merit but does not fully meet PLOS ONE’s publication criteria as it currently stands. Therefore, we invite you to submit a revised version of the manuscript that addresses the points raised during the review process.

We look forward to receiving your revised manuscript.

Kind regards,

Citrawati Dyah Kencono Wungu

Academic Editor

PLOS ONE

Journal Requirements:

Additional Editor Comments:

2. Cite the name in the main text for the name citation, not the number, for example, in line 449 - 451

---

## [Author Response · Author response to Decision Letter 3]

25 Jun 2025

All done! Please let me know if you require any further changes. Thx!

---

## [Editor Report · Decision Letter 3]

30 Jun 2025

Dear Dr. Tyrell,

Thank you for submitting your manuscript to PLOS ONE. After careful consideration, we feel that it has merit but does not fully meet PLOS ONE’s publication criteria as it currently stands. Therefore, we invite you to submit a revised version of the manuscript that addresses the points raised during the review process.

We look forward to receiving your revised manuscript.

Kind regards,

Citrawati Dyah Kencono Wungu

Academic Editor

PLOS ONE

Additional Editor Comments (if provided):

2. Check again the typos in all parts of the manuscript. For instance, in line 297, it is unclear which figure is cited.

3. The reference style is not suited to the journal style. You may check again our guideline: https://journals.plos.org/plosone/s/submission-guidelines#loc-references

---

## [Author Response · Author response to Decision Letter 4]

16 Jul 2025

We would like to thank you for your helpful comments. We have carefully revised the manuscript according to the PLOS ONE style requirements and addressed all the points raised. All modifications are visible using Track Changes in the revised manuscript. Additionally, the figures have been prepared and formatted based on the Preflight Analysis and Conversion Engine (PACE) digital diagnostic requirements.

Editor Comments:

Thank you for the comment. We have carefully revised the manuscript to align with the PLOS ONE style requirements, including file naming conventions, as outlined in the provided templates.

2. Check again the typos in all parts of the manuscript. For instance, in line 297, it is unclear which figure is cited.

We appreciate your attention to this point. We have thoroughly checked the manuscript for typos and clarified the figure reference. The sentence now clearly reads (line 299):

“Fig 2 shows a visual representation of how bias influences the model's decision-making process.”

3. The reference style is not suited to the journal style. You may check again our guideline: https://journals.plos.org/plosone/s/submission-guidelines#loc-references

Thank you for pointing this out. The reference list has been revised to fully comply with the PLOS ONE reference style guidelines.

---

## [Editor Report · Decision Letter 4]

3 Sep 2025

Evaluating the Impact of Sex Bias on AI Models in Musculoskeletal Ultrasound of Joint Recess Distension

PONE-D-24-47665R4

Dear Dr. Pascal,

We’re pleased to inform you that your manuscript has been judged scientifically suitable for publication and will be formally accepted for publication once it meets all outstanding technical requirements. We recommend you to upload high-resolution figures in the copyediting process.

Kind regards,

Citrawati Dyah Kencono Wungu

Academic Editor

PLOS ONE

---

## [Editor Report · Acceptance letter]

PONE-D-24-47665R4

PLOS ONE

Dear Dr. Tyrrell,

I'm pleased to inform you that your manuscript has been deemed suitable for publication in PLOS ONE. Congratulations! Your manuscript is now being handed over to our production team.

Kind regards,

on behalf of

Dr. Citrawati Dyah Kencono Wungu

Academic Editor

PLOS ONE